# PKN1 Kinase: A Key Player in Adipocyte Differentiation and Glucose Metabolism

**DOI:** 10.3390/nu15102414

**Published:** 2023-05-22

**Authors:** Fernando Herrerías-González, Andrée Yeramian, Juan Antonio Baena-Fustegueras, Marta Bueno, Catherine Fleitas, Maricruz de la Fuente, José C. E. Serrano, Ana Granado-Serrano, Maite Santamaría, Nadine Yeramian, Marta Zorzano-Martínez, Conchi Mora, Albert Lecube

**Affiliations:** 1Experimental Surgery Research Group, General and Digestive Surgery Department, Arnau de Vilanova University Hospital, University of Lleida, 25716 Lleida, Spain; fherrerias.lleida.ics@gencat.cat (F.H.-G.); jabaena@vhebron.net (J.A.B.-F.); mcruzdfluente@hotmail.com (M.d.l.F.); maitesantago@gmail.com (M.S.); 2Institut de Recerca Biomèdica Lleida (IRB-LLeida), 25198 Lleida, Spain; mbueno.lleida.ics@gencat.cat (M.B.); conchi.mora@mex.udl.cat (C.M.); 3Department of Experimental Medicine, University of Lleida, 25198 Lleida, Spain; 4Obesity, Diabetes and Metabolism (ODIM) Research Group, Endocrinology and Nutrition Department, Arnau de Vilanova University Hospital, University of Lleida, 25716 Lleida, Spain; 5Biobank Unit, Hospital Universitari Arnau de Vilanova, IRB-Lleida, 25198 Lleida, Spain; 6Department of Biotechnology and Food Science, Faculty of Science, University of Burgos, 09001 Burgos, Spain; nyeramian@ubu.es; 7Immunology Unit, Department of Experimental Medicine, Faculty of Medicine, University of Lleida, 25716 Lleida, Spain

**Keywords:** PKN1, visceral adipose tissue, insulin resistance, type 2 diabetes, adipocyte, glucose metabolism

## Abstract

Adipocyte dysfunction is the driver of obesity and correlates with insulin resistance and the onset of type 2 diabetes. Protein kinase N1 (PKN1) is a serine/threonine kinase that has been shown to contribute to Glut4 translocation to the membrane and glucose transport. Here, we evaluated the role of PKN1 in glucose metabolism under insulin-resistant conditions in primary visceral adipose tissue (VAT) from 31 patients with obesity and in murine 3T3-L1 adipocytes. In addition, in vitro studies in human VAT samples and mouse adipocytes were conducted to investigate the role of PKN1 in the adipogenic maturation process and glucose homeostasis control. We show that insulin-resistant adipocytes present a decrease in PKN1 activation levels compared to nondiabetic control counterparts. We further show that PKN1 controls the adipogenesis process and glucose metabolism. PKN1-silenced adipocytes present a decrease in both differentiation process and glucose uptake, with a concomitant decrease in the expression levels of adipogenic markers, such as PPARγ, FABP4, adiponectin and CEBPα. Altogether, these results point to PKN1 as a regulator of key signaling pathways involved in adipocyte differentiation and as an emerging player of adipocyte insulin responsiveness. These findings may provide new therapeutic approaches for the management of insulin resistance in type 2 diabetes.

## 1. Introduction

Obesity is an increasing public health issue and a well-established risk factor for the development of insulin resistance and type 2 diabetes mellitus (T2DM) [1,2]. The mechanisms linking obesity and insulin resistance have been extensively studied, However, the signaling pathways that promote a dysfunctional adipocyte to develop insulin resistance are not fully elucidated. A dysfunctional adipocyte can play a key role in the development of diabetes, as defects in insulin action in adipocytes may lead to systemic insulin resistance with impaired insulin action in muscle and liver [3].

Insulin is a key regulator of metabolic homeostasis, and it exerts its activity by accelerating the rate of glucose disposal in muscle and adipose tissue. After binding its receptor, insulin activates a sequential cascade that ends up inducing GLUT4 translocation to the cell membrane. The phosphorylated insulin receptor (IR) induces the phosphorylation of insulin receptor substrates (IRSs) on multiple tyrosine residues, which induces the activation of phosphoinositide 3-Kinase (PI3K) and Ras-MAPK pathways. The activation of PI3K will lead to the activation of phosphoinositide-dependent kinase 1 (PDK1) that activates the serine/threonine kinase Akt/PKB. Other kinases such as atypical phosphokinases (PKC), exemplified by PKCζ, can be also activated by PI3K [4] and PDK1 [5].

The protein kinase C-related kinase (PRK) family of serine/threonine kinases is composed of three isoforms: PRK1(PKN1/PKNα), PRK2 (PKN2/PKNγ) and PRK3 (PKN3/PKNβ) [6,7], which are downstream effectors of the Rho family small G proteins. PRK kinases are activated by PDK1 (3-phosphoinositide-dependent kinase), by phosphorylating a threonine residue in their activation loop [8,9]. Rho family GTPases behave as molecular switches for signaling pathways and are best known as orchestrators of actin cytoskeletal rearrangements, with diverse roles in cell migration [10], the cell cycle [11] and the control of vesicular trafficking [12]. Moreover, the ability of PRKs to bind to PDK1 has been shown to be under the control of Rho GTPases [9]. Of note, PKN1 kinase controls different biological processes, such as cell adhesion, cytoskeletal dynamics and tumor cell migration and invasion [10]. PKN1 was also shown to control insulin-stimulated glucose transport [12], as the stable expression of wild type PKN1 in rat adipocytes increases GLUT4 translocation to the plasma membrane and glucose transport [12]. Moreover, PDK1 has been reported to mediate insulin signal to the cytoskeleton [13].

In this paper, we aimed to address the role of PKN1 kinase in obesity-associated insulin resistance, in insulin signaling and in adipocyte biology. Results obtained from visceral adipose tissue (VAT) in humans and from in vitro murine adipocytes show that PKN1 activation levels are lower in insulin-resistant conditions compared to the control. We provide evidence that PKN1 is needed for full adipocyte differentiation and for its function, as PKN1-deficient adipocytes are hypertrophic and present an impaired insulin activation with a reduced capacity in glucose transport. We propose that low activation levels of PKN1 are associated with dysfunctional adipocyte and insulin-resistant adipocyte phenotype.

## 2. Materials and Methods

### 2.1. Ethical Statement

This study was carried out in accordance with the Helsinki Declaration and after obtaining the approval of the Arnau de Vilanova Hospital Research Ethic Committee (CEIC-2167). Samples were obtained from the Biobank with the support of the Xarxa de Bancs de Tumors de Catalunya, sponsored by “Pla Director d’Oncología de Catalunya (XBTC), IRBLleida Biobank (B.0000682)” and “Plataforma Biobancos PT17/0015/0027”. Informed written consent was obtained from patients.

### 2.2. Patient Samples Processing

Human VAT biopsies were obtained from 31 consecutive patients with severe obesity at the time of abdominal bariatric surgery, after an overnight fast. The diagnosis of diabetes was carried out as stated by the American Diabetes Association guidelines [14]. Immediately after excision, the fat tissues were snap frozen in liquid nitrogen and then stored at −80 °C until further processing. Sample lysates were obtained by lysing adipose tissues in NP-40 lysis buffer, supplemented with 1:50 Halt Phosphatase Inhibitor cocktail (Thermo Fisher Scientific, Waltham, MA, USA) and 1:50 Halt Protease inhibitor cocktail EDTA-free (Thermo Fisher Scientific), prior to a 20 min lysis on a wheel at 4 °C and then centrifuged for 10 min at 4 °C at 13,000 rpm to remove all debris. The supernatant was transferred to a new tube and was used to measure protein concentration using the Pierce BCA protein assay. The remaining aliquots were stored at −80 °C until further processing.

### 2.3. Reagents and Antibodies

Biochemical reagents, such as Cytochalasin B (C-6762), 2-deoxyglucose (D-8375), Dexamethasone (D2915), isobutymethylxanthine (I5879), Insulin (I0516), Rosiglitazone (R2408), BCS (Bovine Calf serum, 12133C), BSA (Bovine Serum Albumin, A9647) and Oil Red O (O0625), were all purchased from Sigma-Aldrich (St. Louis, MO, USA). 2-[3H] deoxyglucose was purchased from Perkin Elmer (NET328250UC). All lentiviral plasmids harboring shRNA sequences (mission shRNA) were obtained from Sigma. Antibodies against β-actin (Sigma-Aldrich, St Louis, MO, USA; a5441), α-tubulin (clone B-5-1-2) and total PKN1 were purchased from Sigma-Aldrich. Phospho-IRS-1 (Tyr895), Phospho-PRK1 (Thr774)/PRK2 (Thr816), Phospho-Akt (Ser473) and Phospho-p70 S6 kinase (Thr 389) were purchased from cell signaling. Peroxidase conjugated goat anti-mouse (115-035-003) and goat anti-rabbit (111-035-003) antibodies were obtained from Jackson Immunoresearch (West Grove, PA, USA).

### 2.4. Cell Culture, Adipocyte Differentiation and Cell Area Measurements

Mouse 3T3L1 fibroblasts were obtained from Sigma and cultured in DMEM containing 10% BCS and antibiotics (penicillin and streptomycin). They were seeded on 6-well plates and differentiated into adipocytes, as described previously [15]. Briefly, two days after reaching confluence (termed D0), cells were stimulated for 48 h with DMEM supplemented with 10% fetal bovine serum (FBS), 1 µM Dexamethasone, 0.5 mM isobutymethylxanthine (IBMX) and 10 µg/mL insulin and 2 µM Rosiglitazone (DI2). Two days later, the medium was replaced by DMEM supplemented with 10% fetal bovine serum (FBS) and 10 µg/mL insulin for 48 h. Subsequently, cells were re-fed with DMEM + 10% FBS every two days. Insulin resistance model was induced as described by Hoehn et al. [16]. Briefly, 3T3-L1 adipocytes were incubated with 10nM of insulin, which was added to the culture at 12:00 AM., 16:00 P.M. and 20:00 P.M of the first day and at 08:00 A.M. on the following day. On the following day, cells were washed twice with PBS and then serum starved for two supplemental hours prior to insulin stimulation. For adipocyte cell area measurements, cells were pictured at 20×, and the area was measured using ImageJ software (version 2.1.0/1.53c).

### 2.5. Western Blot Analysis

PKN1 activation levels were assessed via Western blot technique using an anti-Phospho-PRK1 (Thr774)/PRK2 (Thr816) antibody. Cells were washed with cold PBS and either lysed with NP-40 buffer supplemented with Proteases and Phosphatases inhibitors or with SDS lysis buffer (2% SDS, 125 mM Tris–HCL pH6.8), and lysates were subjected to Western blotting, as described in a previous work [17]. Equal amounts of proteins were resolved in 10–15% SDS-PAGE gels and transferred to PVDF membranes (Millipore, Bedford, MA, USA). Membranes were blocked in TBST (20 mM Tris-HCl pH7.4, 150 mM NaCl, 0.1% Tween-20), plus 5% non-fat milk for 1h or with 3% of BSA (bovine serum albumin) to avoid non-specific binding, and were then incubated with the primary antibodies overnight at 4 °C. Membranes were then incubated with peroxidase-coupled anti-mouse or anti-rabbit secondary antibodies for 1h followed by chemiluminescent detection with ECL Advance (Amersham-Pharmacia, Buckinghamshire, UK). For Western blot quantification, the intensity of the lower and upper band was measured using Image Lab software and the ratio was quantified and normalized against the ratio of a sample used as a reference sample.

### 2.6. Oil Red O Staining

Fully differentiated adipocytes (day 14) were stained with Oil Red O to show triglyceride content and visualized by microscopy as described previously [18]. Briefly, fresh Oil Red O stock solution (0.5%) was prepared in isopropanol and filtered. Cells were fixed with PFA 4% for ten minutes and then washed twice with PBS. Meanwhile, a diluted working solution of Oil red O solution was prepared by mixing three parts of Oil Red O stock solution to two parts of H_2_O, allowed to sit for ten minutes, and later filtered. A total of 1.5 mL of the diluted Oil Red O solution was added to cells (p6 well) for 2 h. Cells were then rinsed 3 times with H_2_O and dried and photographed.

### 2.7. 2-[^3^H] Deoxyglucose Uptake Assay in 3T3-L1-Differntiated Adipocytes

3T3-L1-differentiated adipocytes were incubated with serum-free medium for two hours. Following serum starvation, cells were washed twice with PBS, and then were either left untreated, or were stimulated for 30 min at 37 °C with insulin at a concentration of 100 nM. Then, cells were washed twice with warm PBS, and glucose uptake was carried out by adding [H3] 2-deoxyglucose at a concentration of 1 µCi/mL and 100 µM of unlabeled 2-deoxyglucose for 5 min at 37 °C, as previously described [19]. Uptake was terminated through two washes with ice cold PBS, after which cells were lysed with 0.2% SDS buffer. Measurements were made in duplicates. Glucose uptake was detected by scintillation counting. Nonspecific glucose uptake value was determined by adding 20 µM of Cytochalasin B and subtracted from the obtained measurements. Final uptake results were normalized for protein content.

### 2.8. Lentivirus Packaging and Infection and Generation of Stably Transduced 3T3L1 Cells

Lentiviral-based vectors for RNA interference gene silencing were pLKO.1-puro (Sigma-Aldrich), containing either shRNAs or scrambled sequences. PKN1 shRNA lentiviral vectors were purchased from Sigma (Mission shRNA) and were as follows: TRCN0000001485 for shRNA1, TRCN0000001482 for shRNA2, TRCN0000001484 for shRNA3 and TRCN0000012708 for shRNA4. Lentiviruses were produced by co-transfecting HEK 293T packaging cells as described previously [20]. STable 3T3-L1 knockdown-resistant cells were selected with Puromycin (2.5 μg·mL^−1^) and then differentiated to adipocytes.

### 2.9. RNA Isolation and Gene Expression Analysis by RT-PCR

Total RNA was extracted, retrotranscribed and subjected to a quantitative polymerase chain reaction (QPCR), as described elsewhere (ABI Prism 7000HT, Applied Biosystems, Foster City, CA, USA) [21]. Briefly, Trizol was used for total RNA extraction from differentiated adipocytes, and when indicated, genomic DNA was removed using DNase I, RNAse-free (Thermo Scientific, Rockford, IL, USA). A total of 1 µg of RNA was used for the retro transcription, and cDNA was generated using the High-Capacity cDNA Archive Kit (Applied Biosystems, Foster City, CA, USA). cDNA samples were subjected to quantitative polymerase chain reaction (QPCR) (ABI Prism 7000HT, Applied Biosystems). The probes used for the quantification of the genes of interest were the following: Mm00440940-m1 for PPARγ2, Mm00456425-m1 for Adiponectin, Mm00445880-m1 for FABP4 and Mm01265914-s1 for CEBPα. The transcript levels of the amplified genes were normalized in each sample to the β-2 microglobulin (Mm00437762-m1 probe) levels. Samples were assayed in triplicates and the relative expression of mRNA was calculated using the 2^−ΔΔCT^ method.

### 2.10. Statistical Analysis

Statistical analyses were performed using GraphPad Prism 8.0.1 (GraphPad software Inc, La Jolla, CA, USA). Normality was checked using the Kolmogorov–Smirnov (for *n* > 50) or Shapiro–Wilk test (for *n* < 50), followed by the *t*-test (parametric) or Mann–Whitney *U*-test (non-parametric test). Results are presented as mean ± S.E, and *p*-values less than 0.05 were considered significant. The specific statistic test applied in each case is indicated in the figure legend. *p*-values are indicated by asterisks * *p* < 0.05, ** *p* < 0.01 and *** *p* < 0.001.

## 3. Results

### 3.1. PKN1 Activation Levels in VAT of Patients with Obesity Inversely Correlate with T2DM Presence

Main clinical data and pPKN1 activation level in VAT from the 31 patients with severe obesity included in the study are displayed in Table 1, according to the presence of T2DM. Except for fasting plasma glucose (FPG) and HbA1c, both groups had similar age, BMI, triglycerides, and cholesterol levels. The ratio of pPKN1 (Thr 774)/β-actin levels was 60.74% lower in the analyzed VAT from subjects with T2DM compared to the VAT from their non-diabetic counterparts (1.08 ± 0.19 vs. 0.42 ± 0.12; *p* = 0.0175) (Table 1, Figure 1A,B). Moreover, pPKN1 (Thr 774)/β-actin levels inversely correlated with HbA1c (r = −0.533, *p* = 0.02) and FPG (r = −0.454, *p* = 0.01) (Figure 1C).

### 3.2. PKN1 Is Phosphorylated by Insulin Signaling in 3T3-L1 Adipocytes, and This Phosphorylation Is Hindered under Insulin-Resistance Conditions

The decreased activation levels of PKN1 in VAT of patients with T2DM led us to investigate whether insulin may regulate PKN1 activation levels. To test our hypothesis, we conducted a pilot experiment that enabled us to determine the best time point of insulin action on adipocytes, and to assess the activation of downstream targets. Hence, fully differentiated 3T3-L1 adipocytes were treated with 100 nM of Insulin, for up to 24 h, and PKN1 activation levels were assessed with an anti phosho-PKN1 (Thr774) antibody. As shown in Figure 2A, insulin stimulated PKN1 activation. Hence, when treated with 100nM of insulin, 3T3-L1 adipocytes increased PKN1 phosphorylation levels. The insulin induced activation pathway can also be observed with the adaptor protein IRS1, which undergoes phosphorylation on Tyr 895 residue. Furthermore, PKB/Akt activation levels, measured by the phosphorylation of Ser 473, were also increased by insulin treatment. Altogether, these results suggest that the three proteins present a synchronized activation in response to insulin. Moreover, PKN1 and IRS1 present an exquisitely orchestrated activation, as their activation peak is maximal at 10 to 30 min post-insulin treatment and decreases at 3 to 24 h’ post-insulin stimulation. This result suggests that the insulin signaling pathway might involve both proteins, as the early steps of the insulin-triggered activation pathway. In contrast, although PKB activation, i.e., Akt phosphorylation at Ser473, showed maximal phosphorylation in the 10 min following insulin treatment, and its activation was sustained for 24 h (Figure 2A), most likely underscoring the numerous metabolic effects triggered by this Kinase activation. Altogether, these results suggest that PKN1 is activated by insulin and may be involved in regulating insulin-dependent glucose uptake.

To analyze the role of PKN1 activation in response to insulin, we established an experimental model of hyperinsulinemia, mimicking insulin resistance in vitro. As shown in Figure 2B, 3T3-L1 adipocytes incubated under chronic insulin condition and stimulated with insulin presented a significantly lower glucose uptake when compared to their control counterparts (*p* = 0.0286). Lysates from the same wells used for glucose uptake were then processed for Western blotting. As shown in Figure 2C, while insulin induced an increase in PKN1 activation levels (Thr774) in control adipocytes, it induced no change in pPKN1 levels in resistant adipocytes. Moreover, normal adipocytes stimulated with insulin exhibited an increase in IRS1 phosphorylation levels (Tyr 895), PKB/Akt (Ser-473) and mTORC1 substrate p70S6K (ribosomal protein S6 kinase beta-1) activation, while this activation was partially inhibited in insulin resistant adipocytes. Furthermore, while PKN1 total levels remained constant in all analyzed conditions, a positive correlation was found between pPKN1 (Thr774)/β-actin levels and glucose uptake in the analyzed conditions (r = 0.9947, *p* = 0.0053). Together, these results support the hypothesis that PKN1 activation levels correlate with insulin responsiveness in mouse adipocyte.

### 3.3. PKN1 Is Activated during Adipocyte Differentiation and Is Required for Full Adipocyte Differentiation

Insulin resistance that occurs in type 2 diabetes is often linked to adipocyte dysfunction [22]. Over the two past decades, extensive research works have been focused on deciphering the adipocyte differentiation transcriptional cascade [23], and various transcription factors, such as C/EBPα and PPARγ, have been shown to be crucial for both adipogenesis and insulin sensitivity acquisition [24,25]. We thus wondered whether PKN1 activation may be involved in the adipocyte differentiation process. To address this question, we differentiated 3T3-L1 cells to adipocytes and analyzed lysates at different differentiation days: day 0 or pre-adipocytes (NDI, no differentiation induction) and adipocytes at 2, 4 and 6 post-differentiation days. As shown in Figure 3A, while pre-adipocytes showed low levels of total and pPKN1 (Thr 774), these levels increased after two days of differentiation induction (DI2) and were maintained during days 4 and 6 (DI4 and DI6, respectively). This increase in pPKN1 activation levels was accompanied with an increase in AKT and p70S6 kinases phosphorylation levels at Ser 473 and Thr389, respectively. As PKN1 is activated during adipocyte differentiation, we wondered whether its presence was needed for adipocyte full differentiation. To tackle this issue, we developed four different shRNA lentiviral vectors targeting PKN1 mRNA. 3T3-L1 cell lines were transduced with these lentiviral vectors followed by puromycin selection to ensure the stable expression of PKN1 shRNA. As shown by Western blotting in Figure 3B, pre-adipocytes harboring PKN1 shRNA presented a significant decrease in PKN1 protein levels compared to control shRNA. After two weeks of differentiation, PKN1-silenced adipocytes presented a reduced differentiation capacity compared to control shRNA-infected 3T3-L1, as assessed by Oil Red O staining (Figure 3C). Moreover, adipocytes lacking PKN1 presented a statistically significant increased size when compared to their control counterparts with areas of 625.86 ± 13.57 and 591.6 ± 16.41 µm^2^ for control shRNA-infected adipocytes and 868.65 ± 18.84 and 903.9 ± 22.43 µm^2^ for PKN1-shRNA 3 and 4, respectively (*p* < 0.0001) (Figure 3D,E). Altogether, these results suggest that PKN1 is activated during adipogenesis, and its deficiency decreases the number of adipocytes that become hypertrophic and present lipid storage defects compared to their control counterparts (Figure 3D,E). The impaired lipid storage and the size increase observed in PKN1-depleted adipocytes is most probably a direct outcome of defective adipocyte differentiation.

### 3.4. PKN1 Is Required for Full Adipocyte Differentiation

To assess whether the observed phenotype in PKN1-depleted adipocytes is related to the dysregulation of adipogenic process, we analyzed the expression levels of selected transcription factors implicated in adipogenesis. First, we assessed the expression levels of PPARγ, the master regulator of adipocyte differentiation. PPARγ activation is an early event in the early stages of adipogenesis, and its presence is necessary and sufficient for adipogenesis [22]. As shown in Figure 4A, PKN1-depleted adipocytes presented on day 2 of differentiation reduced the levels of PPARγ, ranging between 65% (shRNA #3) to 81% (shRNA #4) compared to the control counterparts. This result indicates that PKN1 is required for adipogenesis initiation. Furthermore, the expression analysis of differentiated adipocyte markers shows that the expression of mature adipocyte genes, such as FABP4, Adiponectin and CEBPα, are significantly downregulated in PKN1-silenced mature adipocytes compared to the control shRNA-transduced counterparts (Figure 4B–D), indicating that PKN1 is essential for the acquisition of a mature adipocyte phenotype. Altogether, our results point to PKN1 as a key regulator of adipocyte differentiation, as it controls the expression levels of various transcription factors involved in early and late adipogenesis.

### 3.5. PKN1 Knockdown Impairs Insulin Signaling and Glucose Uptake in 3T3-L1 Adipocytes

Transcription factors PPARγ and CEBPα have been reported not only to cooperate to activate the full adipogenic program but also to control insulin sensitivity [25]. Based on our results related to morphological and gene expression profile of PKN1-depeleted adipocytes, we next wanted to assess the role of PKN1 in insulin-induced glucose uptake. We thus knocked down PKN1 expression in 3T3-L1 pre-adipocytes using lentiviruses harboring PKN1 shRNA (#4) and then differentiated them to adipocytes. As shown in Figure 4E, PKN1 knockdown significantly reduced insulin-stimulated 2DG (2-Deoxy-d-glucose) uptake in fully differentiated adipocytes (scr adipocytes, 3.051 ± 0.526, vs. PKN1-depleted adipocytes, 1.552 ± 0.115; *p* = 0.019). PKN1-depleted adipocytes presented significantly reduced Akt and p70S6kinase phosphorylation levels (Figure 4F). To avoid misleading results due to the non-specific off-targets effects of shRNA, we tested glucose intake with two other different shRNA sequences against PKN1. As shown in Figure 4G, glucose uptake is significantly reduced in both knockdown cells with a decrease rate of 2DG uptake of 65.49% and 62.73% for shRNA#1 and shRNA#3 transduced adipocytes, respectively (scr adipocytes, 3.549 ± 0.748, vs. PKN1-depleted adipocytes, 1.224 ± 0.209 (#shRNA1) and 1.322 ± 0.235 (#shRNA3), *p* = 0.029 and *p* = 0.024, respectively). Altogether, these results point to PKN1 as a specific regulator of the insulin-mediated glucose uptake pathway. Moreover, the defect in glucose uptake observed in Figure 4G was accompanied by a decrease in insulin regulated AKt phosphorylation (Ser 473) (Figure 4H). Overall, our results indicate that PKN1 is a crucial player that controls adipocyte sensitivity to insulin.

## 4. Discussion

Different protein kinases have been shown to exert regulatory effects controlling various aspects of adipocyte biology [26]. In this paper, we have assessed the role of PKN1 kinase in insulin signaling and in adipocyte biology not only in the experimental model with in vitro murine adipocytes but also in the human VAT. Our results suggest that PKN1 activation decreases in insulin-resistant conditions, impairing the capacity of adipocyte in glucose transport. In addition, we have also provided evidence that PKN1-deficiency impairs early and late adipocyte differentiation, favoring cell hypertrophy and stopping the acquisition of mature adipocyte phenotype.

Visceral adipose tissue is a decisive player in the establishment of T2DM. Functional adipocytes promote insulin sensitivity [27] by controlling the secretion of adipokines and the sequestration of lipids, such as triglycerides stores. Insulin sensitivity is also controlled by adipocyte differentiation, a complex process that involves the activation of various transcription factors, such as PPARγ and CEBPα, which control the expression of genes involved in lipid metabolism and the acquisition of insulin sensitivity, respectively [25,28]. As a result, a dysfunctional adipocyte ends up by presenting an altered glucose metabolism due to a defective insulin pathway, a condition known as insulin resistance [27].

Insulin induces glucose transport by activating a cascade of reactions that start with the binding of insulin to its receptor (IR), followed by the activation of the PI-3 kinase pathway mainly through Akt/PKB cascade and ends in the increasing of the translocation of Glut-4 from intracellular vesicles to the plasma membrane [29]. Akt activity is regulated by phosphorylation at Thr308 in the activation loop and is mediated by PDK1 serine/threonine Kinase (3-phosphoinositide-dependent protein kinase 1) and on Ser473 by the mTORC2 complex (consisting of mTOR, RICTOR, SIN1 and mLST8), respectively. The full activity of Akt depends on both phosphorylation events. Recent reports [30] suggest that under insulin signaling, a complex termed KARATE, formed by GDP-bound RHOA and mTORC2, is responsible for phosphorylating Akt on the Ser 473 residue. Moreover, these authors and others [31] have pointed to posttranslational modifications, such as phosphorylation, and the formation of specific protein complexes, such as the supplemental regulatory mechanism of RHOA activation. Once activated, Akt phosphorylates AS160 (Akt substrate, 160 KDa) and the phosphorylation that suppresses its Rab GTPase-activating protein activity is mandatory for permitting insulin-stimulated GLUT4 translocation to the plasma membrane of adipocytes [32].

The involvement of PKN1 in insulin signaling was first reported by Standaert et al., who showed that increased levels of PKN1 promote glucose transport by increasing Glut-4 glucose transporters translocation to the cell membrane [12]. This translocation is most probably enhanced by the interaction of PKN1 with phospholipase D1 [33], known to regulate Glut-4 transporter containing vesicles translocation to plasma membrane [19]. PKN1 activation is dependent on its interaction with Rho family GTPases that bind to its HR1 domain (homology Region 1) disrupting the auto inhibitory intramolecular interaction of PKN1, relieving the pseudo substrate autoinhibition, leading to its activation [34], while PDK1 activates PKN1 by phosphorylating its threonine residue (Thr 774) [9]. Moreover, the interaction between PKN1 and PDK1 kinases has been shown to be Rho-dependent [9], suggesting that Rho binding to PKN1 allows its access to PDK1. Moreover, Dong et al. have shown that PKN1 participates in actin cytoskeletal reorganization, and that its activation via PDK1 is vital for transducing the signal from the insulin receptor (IR) to the actin cytoskeleton [13], while Yang et al. showed that mTORC2 activates the PKN1 turn motif [35].

Despite all the above-cited available data about the insulin activated pathway, the exact intracellular pathway linking PKN1, insulin signaling, and Glut-4 translocation is still unknown. Moreover, neither the activation state of PKN1 under diabetic conditions nor its role in adipocyte differentiation have been addressed so far. In the present study, we investigated the potential role of PKN1 in adipocytes. Our results show that insulin-resistant adipocytes present a reduced insulin-stimulated PKN1 activity. These results were also confirmed in the VAT from patients with obesity and diabetes that present lower PKN1 phosphorylation levels compared to non-diabetic counterparts. Moreover, PKN1 activation levels are modulated by insulin in a dose-dependent manner, implying a potential role of PKN1 in the insulin-mediated signaling pathway. The results presented in this study demonstrate that PKN1 regulates glucose uptake by controlling AKt/PKB activity. As PDK1 plays a central role in insulin signal transduction pathway by phosphorylating PKB/AKT1, and one possible explanation would be that PDK1-activated PKN1 would lead to PKB/Akt activation, thus propagating the signal to downstream targets. Moreover, PKN1 needs to interact with Rho GTPase for its full activation [33], and as PKN1 is implicated in Akt Ser 473 phosphorylation, one likely explanation would be that PKN1 acts as a downstream target of the mTORC2/RHOA complex, controlling Akt (Ser 473) phosphorylation. Moreover, our data reveal that PKN1 is activated during the adipocyte differentiation process and that its presence is mandatory for a full differentiation. In fact, adipocytes lacking PKN1 are incompletely differentiated, as shown by a decrease in the expression of adipogenic genes, such as PPARγ, FABP4, Adiponectin and CEBPα. Interestingly, these incompletely differentiated adipocytes are hypertrophic and present an impaired insulin activation pathway, with a decrease in pAkt/PKB and p70S6 kinases activation and a reduced glucose uptake compared to their counterparts.

## 5. Conclusions

In summary, we demonstrated a significant role of PKN1 in controlling both adipocyte differentiation and its responsiveness to insulin. PKN1 deficiency leads to a hypertrophic adipocyte with reduced expression levels of key adipogenic factors. Moreover, PKN1-deficient adipocytes present a hampered response to insulin with a decrease in glucose transport coupled to a decrease in the Akt/PKB activation pathway. Taken together, our results suggest that PKN-1 should be considered as a potential new therapeutic target in the treatment of insulin resistance and diabetes.

## Figures and Tables

**Figure 1 nutrients-15-02414-f001:**
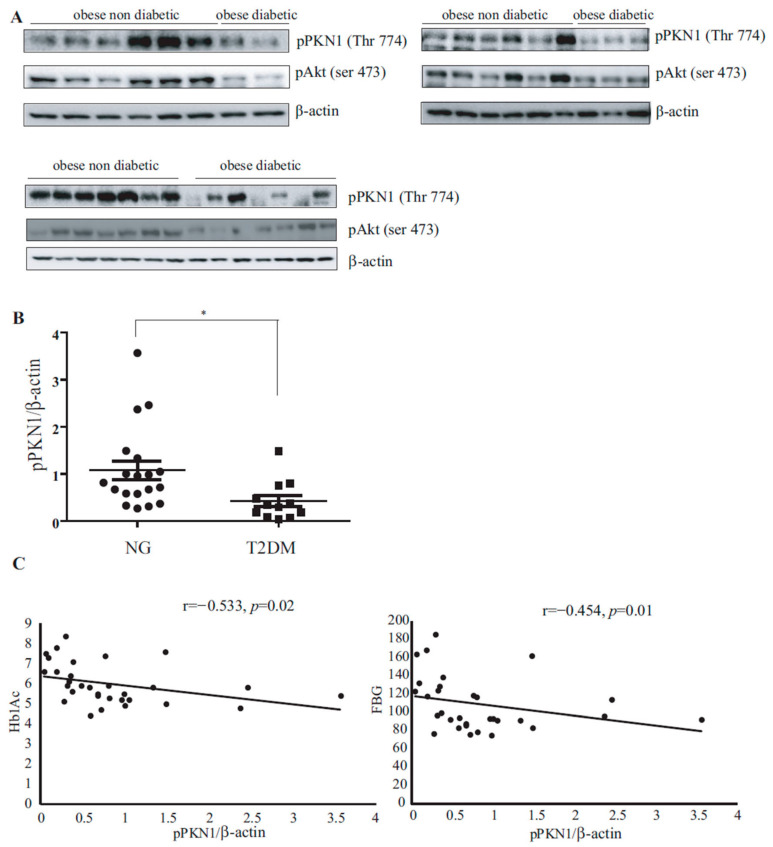
PKN1 and Akt phosphorylation levels in visceral adipose tissue (VAT) samples. (**A**). VAT samples obtained after bariatric surgery from 19 patients without diabetes and 12 patients with type 2 diabetes were analyzed via Western blot for PKN1 and Akt activation levels. Membranes were immunoblotted with Phospho-PKN1 (Thr774) and pAkt (Ser473). (**B**). Statistical analysis of p-PKN1(Thr774)/β-actin ratio levels in VAT of patients with severe obesity with and without type 2 diabetes was performed using Mann–Whitney *U*-test and a shows significant difference in phosphorylation intensity between both groups (* *p* = 0.0112). (**C**). **Left panel:** Spearman’s correlation test between the p-PKN1(Thr774)/β-actin ratio and Hb1Ac shows an inverse correlation between these two variables in the VAT of all analyzed subjects (r = −0.533, *p* = 0.02). (**C**). **Right panel:** Spearman’s correlation test between p-PKN1(Thr774)/β-actin ratio and FBG shows an inverse correlation between these two variables (r = −0.454, *p* = 0.01) in the VAT of all analyzed subjects.

**Figure 2 nutrients-15-02414-f002:**
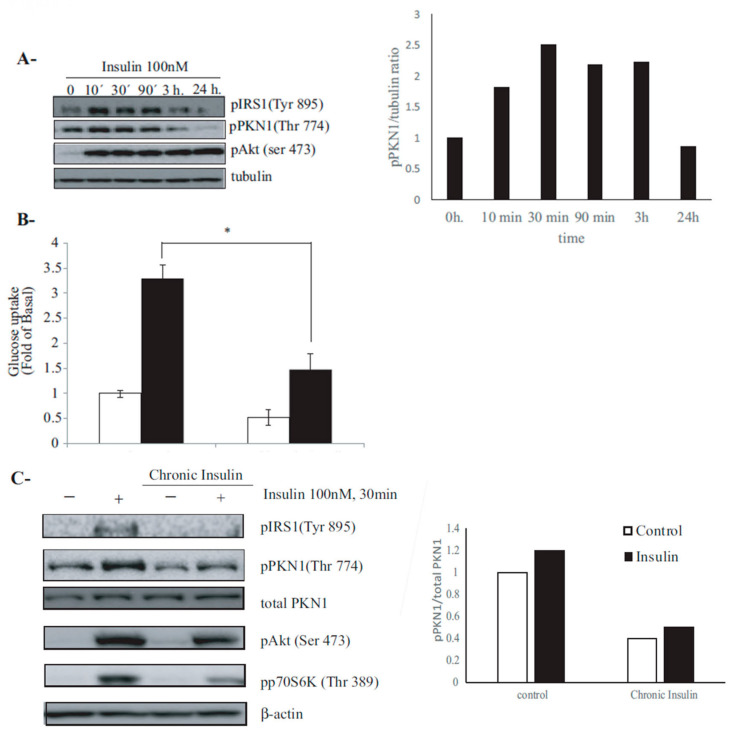
Insulin-stimulated PKN1 phosphorylation is impaired under insulin-resistance conditions in 3T3-L1 adipocytes. (**A**). Fully differentiated 3T3-L1 adipocytes were stimulated for up to 24 h with either 10 or 100nM of insulin. Representative Western blots show a dose-dependent increase in pIRS1 (Tyr895), pPKN1 (Thr774) and p Akt (Ser 473). Tubulin was used as a loading control. pPKN1/tubulin ratios in this pilot experiment are plotted as a function of time. (**B**). Control and insulin-resistant (induced by chronic insulin, 10 nM for 24 h) 3T3-L1 adipocytes were washed, serum starved for 2 h, stimulated with 100 nM of insulin for 30 min and then assessed for [3H]2-deoxyglucose uptake. Results are displayed as mean ± SEM of three independent experiments. Control cells present significantly higher glucose uptake rate compared to insulin-resistant (CI) counterparts (Mann–Whitney U-test, * *p* = 0.0286). (**C**). Cell lysates corresponding to control and insulin-resistant 3T3-L1 adipocytes from the for [3H]2-deoxyglucose uptake experiment (**C**) were immunoblotted using pIRS1 (Tyr895), pPKN1 (Thr774), pAkt (Ser 473) and pP70S6K (Thr389) antibodies. β-actin was used as a loading control. The densitometric analysis of pPKN1/total PKN1 ratios in 3T3-L1 adipocytes were subject to the above-cited conditions. Western blots of cell lysates corresponding to experimental duplication are shown in Appendix A.

**Figure 3 nutrients-15-02414-f003:**
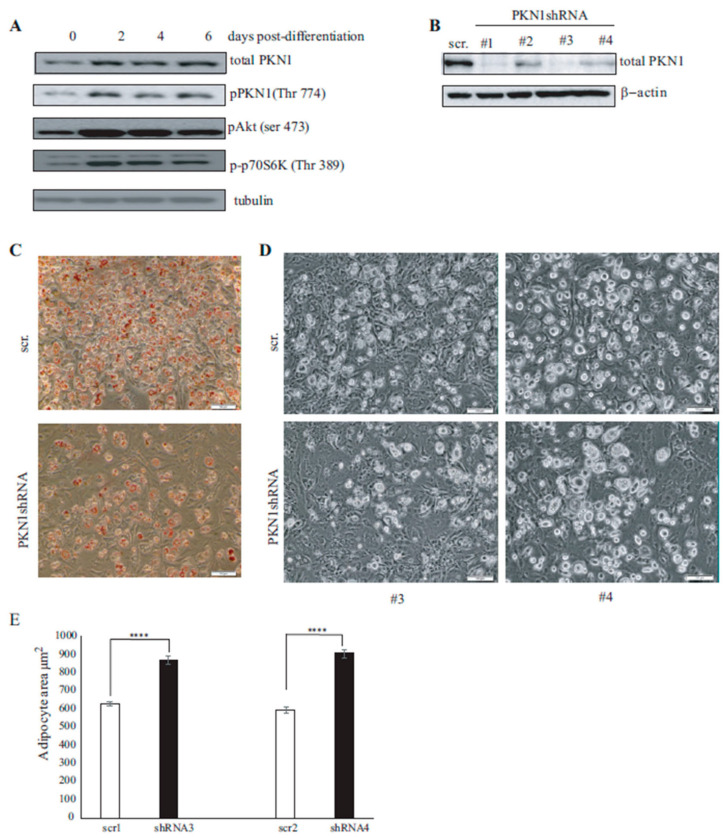
Time-dependent activation of PKN1 during 3T3-L1 adipocyte differentiation: (**A**) PKN1-silenced 3T3-L1 pre-adipocytes were differentiated, and cell lysates of days 2, 4 and 6 of differentiation (D2, D4 and D6) were analyzed via Western blot for total PKN1, pPKN1 (Thr774), pAkt (Ser 473) and pP70S6K (Thr389). Tubulin was used as loading control. (**B**) 3T3-L1 pre-adipocytes were transduced with lentiviral vectors harboring four different shRNA. Cell lysates analyzed by Western blot show the significant downregulation of PKN1 levels in PKN1-shRNA-transduced cells compared to the control shRNA counterpart. β-actin was used as a loading control. (**C**) Oil Red O staining for differentiated 3T3-L1 adipocytes infected with control shRNA and PKN1 shRNA lentiviruses. (**D**) Microscopic phase-contrast images of 3T3-L1 adipocytes infected with two different control shRNA lentivirus (scr1 and scr2) (upper panel) and two different PKN1shRNA lentivirus (#3 and #4 shRNA) (lower panel). Scale bars = 100 µm. (**E**) Quantification of the adipocyte area for each condition of the data in D. Results are represented as mean ± S.E.M from three independent experiments using the Mann–Whitney test, *p* < 0.0001, **** *p* < 0.0001.

**Figure 4 nutrients-15-02414-f004:**
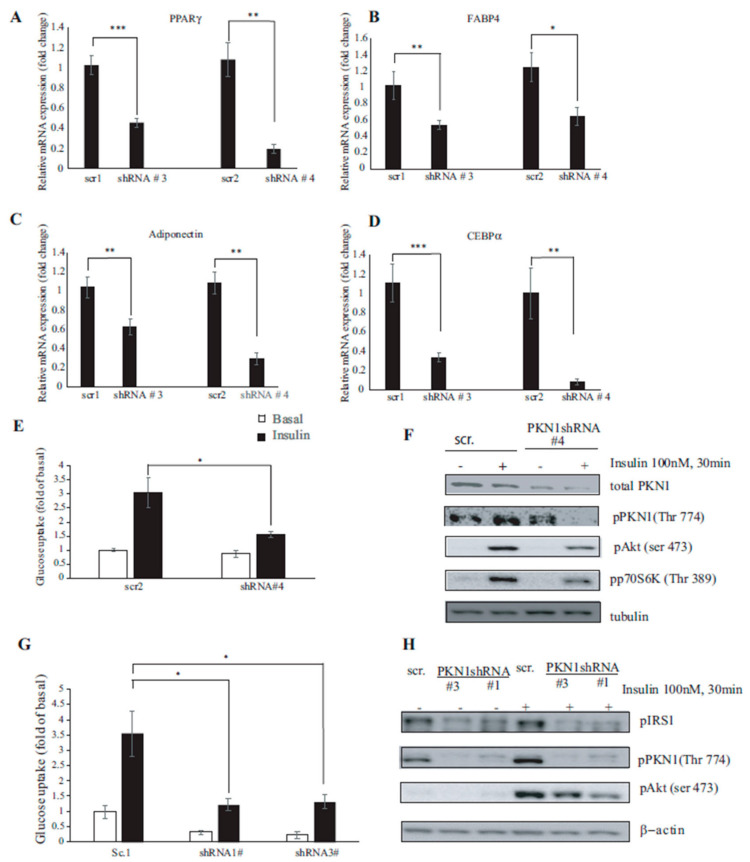
PKN1 reduction in differentiating 3T3-L1 adipocytes leads to a decrease in transcript levels of adipogenic markers and inhibits insulin-stimulated glucose uptake and insulin signaling in 3T3-L1 adipocytes. (**A**) 3T3-L1 pre-adipocytes transduced with lentiviral vector harboring either control (scr1, scr2) or PKN1 shRNA (#3 or #4 shRNA) were differentiated, and the expression levels of early adipogenic marker PPARγ was assessed on day 2 of differentiation using quantitative reverse transcriptase PCR (qRT-PCR). (**B**) FABP4 (fatty acid binding protein 4) levels were measured by qRT-PCR in fully differentiated adipocytes (at day 7 of differentiation). (**C**) Adiponectin and (**D**) CEBPα mRNA relative levels were assessed in fully differentiated adipocytes (day 7) for the indicated conditions. In all experiments, transcript levels were expressed relative to β-2 microglobulin housekeeping gene and values were normalized using control shRNA-transduced adipocytes differentiated for the same period. Results are represented as mean ± S.E.M of three independent experiments using Mann–Whitney *U*-test, * *p* < 0.05; ** *p* < 0.01; *** *p* < 0.001. (**E**) 3T3-L1 pre-adipocytes were transduced with lentiviral vectors harboring either scr2 or shRNA4 to PKN1 oligos. Once differentiated, adipocytes were with or without insulin for 30 min and subjected to 2-DG uptake assay. Results are represented as the mean ± S.E. of three independent experiments using Student’s *t*-test, * *p* < 0.05: ** *p* < 0.01. (**F**) Cell lysates of cells used for 2DG uptake were quantified and subjected to SDS-PAGE. Western blotting was performed using antibodies against total PKN1, pPKN1 (Thr774), pAkt (Ser 473) and pP70S6K (Thr 389). Tubulin was used as a loading control. The blot shown is representative of three independent experiments. (**G**) 3T3-L1 pre-adipocytes were transduced with lentiviral vectors harboring either scramble or different PKN1 shRNA oligos (shRNA#1, shRNA#3). Differentiated adipocytes were subjected to 2DG uptake. Results are represented as the mean ± S.E. of two independent experiments using Student’s *t*-test, * *p* < 0.05: ** *p* < 0.01. (**H**) Western blot of cell lysates from C was performed using antibodies against total pPKN1 (Thr774) and pAkt (Ser 473). Tubulin was used as a loading control.

**Table 1 nutrients-15-02414-t001:** Clinical and metabolic characteristics of patients with obesity according to the presence of type 2 diabetes.

	All	Type 2 Diabetes	Non-Type 2 Diabetes	*p*
*n*	31	12	19	
Age (years)	44.6 ± 11.1	50.2 ± 9.9	40.6 ± 10.4	0.015
BMI (kg/m^2^)	43.6 ± 5.7	42.4 ± 5.1	44.5 ± 6.0	0.323
FPG (mmol/L)	110.6 ± 28.8	137.6 ± 23.9	91.1 ± 10.2	<0.001
Hb1Ac (%)	6.0 ± 1.0	7.0 ± 0.8	5.3 ± 0.3	<0.001
Tryglicerides (mg/dL)	133.2 ± 54.0	151.3 ± 67.92	123.8 ± 43.9	0.175
LDL-cholesterol (mg/dL)	167.2 ± 30.0	108.5 ± 30.7	106.4 ± 30.4	0.860
HDL-cholesterol (mg/dL)	46.6 ± 11.3	42.0 ± 8.9	49.7 ± 11.9	0.067
Total cholesterol (mg/dL)	178.1 ± 33.3	177.9 ± 32.1	178.3 ± 35.1	0.973
pPKN1/β-actin ratio	0.82 ± 0.14	0.42 ± 0.12	1.08 ± 0.19	0.017

Age and clinical parameters are expressed as the mean ± standard deviation. pPKN1/β-actin ratio is expressed as the mean ± standard error of the mean. BMI: body mass index. FPG: fasting plasma glucose. LDL: low density lipoproteins; HDL: high density lipoprotein; pPKN1: phosphorated protein kinase N1.

## Data Availability

The data supporting these findings are included in the original manuscript. Further inquiries related to the results can be addressed to the corresponding author.

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
