# Peer review of "PKN1 Kinase: A Key Player in Adipocyte Differentiation and Glucose Metabolism"

_nutrients, 2023, doi:10.3390/nu15102414_

Round 1

Reviewer 1 Report

In this study, the authors investigated the role of Protein kinase N1 (PKN1) in glucose metabolism under insulin-resistant conditions in primary visceral adipose tissue (VAT) from patients with obesity and murine 3T3-L1 adipocytes. They observed that insulin-resistant adipocytes displayed decreased PKN1 activation levels compared to non-diabetic control counterparts. The authors also demonstrated that PKN1-silenced adipocytes exhibited reduced differentiation and glucose uptake, along with decreased expression levels of adipogenic markers. They propose that PKN1 regulates key signaling pathways involved in adipocyte differentiation and insulin responsiveness. While the study presents interesting findings, the data in its current form is unconvincing and warrants further investigation. The existing data does not sufficiently support the claims that insulin directly activates PKN1 and that PKN1 is directly involved in adipocyte differentiation. My detailed comments are as follows:

Figure 2A: In my view, the impact of insulin on PKN1 activation is mild. Particularly at a concentration of 10 nM insulin, the difference in pPKN1 levels between the control and insulin-treated groups seems negligible, as the loading volume is less in the control sample compared to insulin-treated samples. The authors should provide quantification for this data. To confirm if insulin stimulates PKN1 activation in a dose-dependent manner, the authors should examine pPKN1 levels at varying insulin dosages (e.g., 10 nM, 20 nM, 40 nM, 80 nM) at a specific time point. Additionally, the authors should explain why pPKN1 and pIRS1 levels decrease significantly upon treatment with 100 nM insulin for 3h and 24h. Notably, the remaining amount of PKN1 protein at 3h and 24h is even lower than the control. This decrease is not observed upon treatment with 10 nM insulin.

Figure 2C: The activation of PKN1 upon insulin treatment (lane 1 and lane 2) is discernible in this figure, as the total PKN1 level in the insulin-stimulated group is slightly higher than that in the control group.

Figure 3A: Both total and phospho PKN1 levels are increased, making it difficult to determine if the increased p-PKN1 results from enhanced phosphorylation or an increased amount of PKN1 protein. Also, the tubulin levels are marginally higher in differentiation induction samples compared to the control sample.

Figure 4: The authors should conduct a PKN1 rescue experiment to assess whether PKN1 is directly involved in adipocyte differentiation. An RNAi-resistant PKN1 should be included in the depletion experiment to determine if its expression can restore adipocyte differentiation, mRNA levels of adipocyte genes, and glucose uptake.

Some sentences should be revised to make it easy to understand.

Reviewer 2 Report

The authors have performed studies on human visceral adipose tissue and mouse adipocytes. Using appropriate techniques they have obtained evidence that supports their conclusion that PKN1 kinase is involved in adipocyte differentiation and glucose metabolism. The discussion might be enhanced if the authors contrasted their work with the paper of M.C. Loffler et al. entitled Protein kinase D1 deletion in adipocytes enhances energy dissipation and protects against adiposity (EMBO J. 37 e99182 2018). That paper presents evidence that a different protein kinase may have regulatory effects that in some respects oppose the action of PKN1 kinase.

The following comments are offered:

1.     At the beginning of line 54, change “pokinases” to “phokinases”.

2.     In the legend to Table 1, it is not clear which means are ± standard deviation and which means are ± standard error of the mean.

Author Response

Response to Reviewer 2 Comments

Point 1: The authors have performed studies on human visceral adipose tissue and mouse adipocytes. Using appropriate techniques, they have obtained evidence that supports their conclusion that PKN1 kinase is involved in adipocyte differentiation and glucose metabolism. The discussion might be enhanced if the authors contrasted their work with the paper of M.C. Loffler et al. entitled Protein kinase D1 deletion in adipocytes enhances energy dissipation and protects against adiposity (EMBO J. 37 e99182 2018). That paper presents evidence that a different protein kinase may have regulatory effects that in some respects oppose the action of PKN1 kinase.

Response 1:  The authors appreciate this comment from Referee 2, and accordingly we have included in our discussion the role of other kinases, such as Protein kinase D1, in controlling adiposity, and the above-mentioned reference.

Point 2: The following comments are offered:

  1. At the beginning of line 54, change “pokinases” to “phokinases”.
  2. In the legend to Table 1, it is not clear which means are ± standard deviation and which means are ± standard error of the mean.

Response 2: Thank you for your comments. In response to your comments, we have:

 -corrected the word phosphokinases at the beginning of line 54.

 -In the legend of Table 1, we have specified that age and clinical parameters are expressed as means ± standard deviation, while pPKN1/β-actin ratio is expressed as the mean ± standard error of the mean.

Reviewer 3 Report

The authors have studied the role of protein kinase N1 (PKN1) in adipocyte differentiation and glucose metabolism. They first show that patients with obesity and type 2 diabetes have low levels of active PKN1 (pPKN1) and then using cell culture demonstrate a role for PKN1 in adipocyte differentiation and glucose metabolism. The study is interesting and likely identifies a novel signaling molecule in adipocyte biology. However, the following concerns need to be addressed:

It is unclear how many times experiments were repeated. It is critical to show that results are reproducible, and most experiments should be done at least three times and data should be shown with standard error and statistical analyses. Protein analysis figures should include the blot and quantitative measurement of the bands (e.g., phospho-PKN1/total PKN1) as a graph or table with standard error and significance. Additionally, the fold difference between control and treated will allow the reviewers (and authors) to assess the change due to activation or inhibition.

Fig. 1 – please show Western blot data for total PKN1 as the variation between patients for pPKN1 could be due to different levels of PKN1.

Fig. 2 – see comment above for protein analysis.

Fig. 3 – see comment above for protein analysis.

Fig. 3A – it is unclear if the difference in pPKN1 levels is due to increased activation or increased levels of total PKN1. Quantitative data will help differentiate.

Fig. 3B – in Fig. 3E two different control shRNAs (scr1 and scr2) are used but in 3B only one scr is shown. Please show the effect of both scr on total PKN1.

Fig. 4 – see comment above for protein analysis.

Fig. 4F and H – IRS1 and Akt are upstream of PKN1. Why are they inhibited by PKN1 shRNA?

Figure 5 is missing.

Section 2.4, lines 119 and 120 – the authors state that insulin resistance was induced as described by Hoehn et al 16. However, the reference is not included in the citations.

Section 2.7, lines 159 and 160 – where is the data for glucose uptake in cytochalasin treated cells?

Section 2.9, lines 184 and 186 – these lines are repeated.

Fig 1B X-axis legend is not visible. Most figures can be decreased in size.

Fig 3 legend – what is pre-scheme 3?

There are several formatting errors that should be corrected.

There are minor formatting and grammatical errors, which should be corrected.

Author Response

Response to Reviewer 3 Comments

The authors have studied the role of protein kinase N1 (PKN1) in adipocyte differentiation and glucose metabolism. They first show that patients with obesity and type 2 diabetes have low levels of active PKN1 (pPKN1) and then using cell culture demonstrate a role for PKN1 in adipocyte differentiation and glucose metabolism. The study is interesting and likely identifies a novel signaling molecule in adipocyte biology. However, the following concerns need to be addressed:

It is unclear how many times experiments were repeated. It is critical to show that results are reproducible, and most experiments should be done at least three times and data should be shown with standard error and statistical analyses. Protein analysis figures should include the blot and quantitative measurement of the bands (e.g., phospho-PKN1/total PKN1) as a graph or table with standard error and significance. Additionally, the fold difference between control and treated will allow the reviewers (and authors) to assess the change due to activation or inhibition.

Response: We very much appreciate the comments of Referee 3 to improve the manuscript, and we would like to clarify all experiments have been repeated at least three times (we have specified this in the corresponding Figure legends in the revised version of the manuscript). We have also included additional results (supplementary figures 1A and 1B) that were not included in the original paper. In figure 1A we have included the triplicate experiments of glucose uptake, while in figure 1B, we have tested scr1 and a different shRNA from the ones cited in the manuscript (this shRNA was not included because the vector did not contain a sequence coding for puromycin resistance gene and would not enable a selection of the transduced clones). We have included quantitative analysis of pPKN1/total PKN1 or alternatively pPKN1/tubulin.

Fig. 1 – please show Western blot data for total PKN1 as the variation between patients for pPKN1 could be due to different levels of PKN1.

Response 1: As Referee 3 suggests, it would have been interesting to compare total and pPKN1 levels in patient samples used in Figure 1. However, sample amount was limited, since samples were obtained from the patients’ visceral adipose tissue. Moreover, PKN1 and pPKN1 could not be detected in the same blots, as both antibodies are rabbit antibodies (no other antibodies were available), and therefore could not be blotted in the same membrane. Thus, due to limited sample size and the technical difficulties cited above (antibody detection), we were not able to conduct another Western Blot for total PKN1 determination.

Fig. 2 – see comment above for protein analysis.

We have quantified the ratio of pPKN1/tubulin in the Western 2A, and pPKN1/total PKN1 in Western 2C. Insulin stimulation increases the ratio pPKN1/tubulin, that peaks 30 min post-stimulation. Most probably, this increase in PKN1 phosphorylation levels indicates intrinsic kinase activity as suggested by other reports (2).

Fig. 3 – see comment above for protein analysis.

Fig. 3A – it is unclear if the difference in pPKN1 levels is due to increased activation or increased levels of total PKN1. Quantitative data will help differentiate.

Response 2: As suggested by the Referee 3, and as stated above, the levels of total PKN1 seem to correlate with pPKN1 levels. These data further confirm the existence of an intrinsic autophosphorylation activation loop.

Fig. 3B – in Fig. 3E two different control shRNAs (scr1 and scr2) are used but in 3B only one scr is shown. Please show the effect of both scr on total PKN1.

Response 3: We thank Referee 3 for pointing this out. In fact, scr1 harbored in the construct a sequence encoding GFP, and was used to monitor transduction efficiency by visualizing the cells on fluorescent microscope. The effect of scr1 on total PKN1 levels is shown in supplemental figure 1B.

Fig. 4 – see comment above for protein analysis.

Fig. 4F and H – IRS1 and Akt are upstream of PKN1. Why are they inhibited by PKN1 shRNA?

Response: This is a well taken point. As shown in figure 5 Akt is a downstream target of PKN1, and targeting PKN1 decreases Akt phosphorylation levels. As for IRS1, our data point to the existence of a possible feedback mechanism between PKN1 and IRS1.

Figure 5 is missing.

Response: We apologize, and we have uploaded the figure 5, which corresponds to the graphical abstract.

Section 2.4, lines 119 and 120 – the authors state that insulin resistance was induced as described by Hoehn et al 16. However, the reference is not included in the citations.

Response: Thank you. We have included the corresponding reference in the draft.

Section 2.7, lines 159 and 160 – where is the data for glucose uptake in cytochalasin treated cells?

Cytochalasin treated cells were used to measure nonspecific glucose uptake in each experiment. The value corresponding to the media of the triplicates is subtracted from the obtained measurements in each experiment.

Section 2.9, lines 184 and 186 – these lines are repeated.

We thank the referee for his comment and corrected the error in the manuscript.

Fig 1B X-axis legend is not visible. Most figures can be decreased in size.

Response: We have uploaded the figures in a reduced size and corrected Fig 1B axis legend position.

Fig 3 legend – what is pre-scheme 3? 

There are several formatting errors that should be corrected.

Comments on the Quality of English Language

There are minor formatting and grammatical errors, which should be corrected.

Response: We apologize for these errors. Pre-scheme 3 words were removed as it was an error. We thank the reviewer for this comment.

We have revised the draft and corrected grammatical errors.

REFERENCES

  1. Taniguchi CM, Emanuelli B, Kahn CR. Critical nodes in signalling pathways: insights into insulin action. Nat Rev Mol Cell Biol. 2006;7(2):85-96.
  2. Palmer RH, Ridden J, Parker PJ. Cloning and expression patterns of two members of a novel protein-kinase-C-related kinase family. Eur J Biochem. 1995;227(1-2):344-51.

Round 2

Reviewer 1 Report

Thank the authors for the endeavor contributing to a great improvement of this manuscript. I highly recommend this manuscript for further publication.